# Evaluation of the Antihistamine and Anti-Inflammatory Effects of a Nutraceutical Blend Based on Quercetin, *Perilla frutescens*, *Boswellia serrata*, *Blackcurrant*, *Parthenium*, *Helichrysum*, *Lactobacillus acidophilus* and *Bifidobacterium animalis* Through In Vitro and In Vivo Approaches—Preliminary Data

**DOI:** 10.3390/cimb47110965

**Published:** 2025-11-20

**Authors:** Simonetta Masieri, Francesco Frati, Giulio Torello, Marianna Colasante, Marta Scquizzato, Carlo Cavaliere

**Affiliations:** 1Department of Oral and Maxillofacial Sciences, Sapienza University, 00161 Rome, Italy; simonetta.masieri@uniroma1.it; 2Clinic of Otolaryngology (ENT), University of Perugia, 06123 Perugia, Italy; francescofrati92@hotmail.it; 3TL Pharma Consulting, 65129 Pescara, Italy; giulio.torello@tlpharmaconsulting.it (G.T.); ricerca@tlpharmaconsulting.it (M.C.); 4Science Department, Global Pharmacies Partner, 20123, Milano, Italy; 5Department of Sense Organs, Sapienza University, 00161 Rome, Italy; carlo.cavaliere@uniroma1.it

**Keywords:** antihistamines, side effects, natural products

## Abstract

Respiratory and food allergy conditions are increasing internationally and the most commonly used drugs in these conditions are antihistamines, products that can interfere as histamine receptor antagonists. In accordance with the need to test new principals capable of developing fewer side effects, we preliminarily studied the therapeutic antihistamine effect in vitro and in vivo of an innovative nutraceutical blend based on Quercetin, *Perilla frutescens*, *Boswellia serrata*, *Blackcurrant*, *Parthenium*, *Helichrysum*, *Lactobacillus acidophilus* and *Bifidobacterium animalis.* The in vitro test demonstrated the interaction between the examined mixture and a rat leukemia cell line (RBL-2H3) widely used as a model simulating mast cells in immunological and allergological studies; this pre-clinical test demonstrated a statistically significant reduction in cell histamine degranulation (about 30%). The in vivo test demonstrated instead that the mixture interferes up to 30% in the development of histamine wheal. In addition, during the in vitro test, we also tested the effect of the mixture on allergic inflammation, so we evaluated the interference of the mixture on TNF alpha levels, determining a reduction in tested concentrations of about 13%.

## 1. Introduction

Allergies, commonly called type I hypersensitivity, results from a dysfunctional immunological state in which foreign allergens, harmless to most individuals, provoke a hyper-reactive immune response [1]; allergic disease is the result of the interaction between genes and the environment and manifests itself in the form of various diseases such as allergic rhinitis, allergic asthma, atopic dermatitis, food allergy, and eczema [2], which affect up to 30% of people in Western countries [3]. In particular, allergic rhinitis affects 14% of adults and 13% of children in the United States [4]; globally, 300 million people suffer from asthma [5] and, in high-income countries, atopic dermatitis affects 20% of children and 10% of adults [6]. IgE-dependent allergic reactions involve the process of sensitization to the allergen, activation of Th2 lymphocytes and production of allergen-specific IgE [7]. An allergen crosses the body’s barriers through mucosal surfaces and is taken up by an antigen-presenting cell (APC), degraded intracellularly into peptide fragments and presented on MHC class II molecules. After presenting the allergenic peptide to a CD4+ or helper T cell (Th2), these cells acquire an effector function that stimulates naive B cells to differentiate into IgE effector plasma cells [8]. The isotype switch of B cells leads to the production of allergen-specific IgE, which binds FcεRI on mast cells and basophils. A second encounter with the allergen results in its binding to the IgE already present, determining the release of preformed mediators such as histamine [3] and increased synthesis of many cytokines, chemokines and growth factors. This release of mediators from mast cells contributes to the acute signs and symptoms associated with early-phase allergic reactions such as vasodilation, increased vascular permeability, contraction of bronchial smooth muscle, increased mucus secretion and stimulation of nociceptors of sensory nerves in the nose, skin and airways. Mast cells also contribute to the transition to the late phase reaction by promoting an influx of inflammatory leukocytes and regulating adhesion molecules on vascular endothelial cells [9]. TNFalpha is involved in early and late stages of allergy and plays an important dual role in maintaining immune homeostasis and promoting disease development [10]. TNF-α is a pro-inflammatory cytokine that is released in allergic responses by mast cells and macrophages through IgE-dependent mechanisms. TNF-α is closely involved in allergic rhinitis, from its onset to its development: TNF-α potentiates allergic sensitization, pretreatment with TNF-α prior to allergen sensitization increases antigen-specific IgE antibody responses after allergen exposure, and TNF-α has an important role in the expression of adhesion molecules that induce transendothelial migration of eosinophils [11].

The curative approach for allergies involves eliminating or reducing exposure to allergens as a first step, followed by traditional pharmacotherapy that commonly used antihistamines and corticosteroids; lastly, allergies can be treated, in selected cases, with specific allergen immunotherapy in accordance with ARIA guidelines and specific disease conditions [12,13]. The increasing prevalence of allergies has put anti-allergy drugs at the forefront of the commercial drug market, which is expected to grow rapidly from 2021 to 2029 with valuations at USD 234.59 million in 2021 and USD 476.08 million by 2029 [14]. Histamine exerts its various biological effects through four types of receptors: H1, H2, H3 and H4, all of which are G-protein-coupled. The allergic response is mainly mediated by the H 1 receptor subtype, which is why H 1 receptor antagonists are mainly used in clinical practice as anti-allergy drugs [15]. Antihistamines are drugs that have been in use for more than 70 years with pharmacological characteristics that have evolved over time from first-generation antihistamines to second-generation antihistamines [16]. First-generation antihistamines had a certain degree of central toxicity through crossing the blood–brain barrier and their central anticholinergic effects [17]. Second-generation non-sedating antihistamines are considered the first-line treatment, although they may present side effects such as drowsiness, dry mouth, rash or fatigue [18] as their selectivity decreases at higher dosages and concentrations, affecting the cholinergic, serotonergic and catecholaminergic systems [19]. Some antihistamines, such as terfenadine and astemizole, can also cause cardiac side effects [20]. Since the 1990s there have been reports of syncope and torsade de pointes with these non-sedating antihistamines, as they cause qt prolongation and torsade de pointes, particularly in the case of overdose [21]. Furthermore, terfenadine has also been shown to inhibit the activity of G-protein-gated inwardly rectifying K+ (GIRK) channels, which regulate the excitability of neurons and cardiomyocytes [22]. The known sedative properties of this drug category (drowsiness, impaired performance, etc.) are the result of occupation of the H1RO receptor (brain H 1 receptor) and because of this, the antihistamine sedative potential is divided into non-sedative (<20%), less sedative (20–50%) and sedative (≥50%) [23]. Second-generation antihistamines rarely cross the blood–brain barrier; however, ketotifen and cetirizine still have a relatively mild central sedative effect. Cetirizine still penetrates the brain to a small extent, resulting in an H1RO D occupancy of 26.0% and causing a certain degree of drowsiness [24]. Furthermore, brain penetration of orally administered cetirizine was found to be dose dependent. Cetirizine 10 mg, with its low H1RO and thus minimal sedation, could be used more safely than cetirizine 20 mg for the treatment of various allergic disorders [25].

Conventional therapies can present side effects associated with their routine use and do not always provide a complete resolution of symptoms, so adjuvant, natural and novel therapies targeting the triggering mechanisms of the allergic response are needed [6]. Different plants have been the subject of several studies supporting their antihistamine and anti-inflammatory effects. In this study, we selected a combination of these plants and specific probiotics—*Boswellia serrata*, Quercetin, *Blackcurrant*, *Helichrysum*, *Perilla frutescens*, *Parthenium*, *Lactobacillus acidophilus* and *Bifidobacterium animalis*—to optimize these effects. This selection was not random and each ingredient was supported by extensive scientific literature and known anti-allergic properties. *Boswellia* is a tree of the *Burseraceae* family from which a rubbery oleo-resin consisting of 30–60% resin, 5–10% essential oils soluble in organic solvents and polysaccharides is extracted through an incision made in the trunk [26]. The dry Boswellia extract used in the mixture was a dry extract of *Boswellia serrata* resin, titrated at 75% total boswellic acid and 10% acetyl-11-keto-β-boswellic acid, marketed under the registered name AKBAMAX^®^. Boswellic acids (BA) are pentacyclic triterpene acids on which in vitro and in vivo studies have been performed that show that it causes an inhibition of the synthesis of the pro-inflammatory enzyme 5-lipoxygenase (5-LO), 5-hydroxyhexyeicosatetraenoic acid (5-HETE) and leukotriene B4 (LTB-4), which cause bronchoconstriction, chemotaxis and increased vascular permeability [26]. In addition, the molecular anti-inflammatory mechanism of boswellic acids is also based on the inhibition of inflammatory factors and/or pathways such as prostaglandins (PGs), histamines, leukotrienes and interferons (IFNs), and on the positive regulation of oxygen free radicals. The multiple anti-inflammatory mechanisms have enabled the successful treatment of various diseases (also allergic in nature) using Boswellia extracts [27].

In the mixture under examination, the quercetin used was granular with a minimum degree of 95% purity obtained from *Sophora japonica* L. Quercetin is a natural flavonoid polyphenol found in some fruits and vegetables with high potential benefits: anti-allergic functions and inhibition of histamine production and pro-inflammatory mediators. There are several studies highlighting this characteristic in vitro and in vivo in allergic asthma, allergic rhinitis and atopic dermatitis due to quercetin’s ability to regulate Th1/Th2 balance in a mouse model of asthma, reduce the release of antigen-specific IgE antibodies by B cells and inhibit sneezing in allergic rhinitis [28,29,30,31]. Its anti-oxidative capacity is mediated by the molecular structure of quercetin itself, which enables the delocalization of electrons from the ring, resulting in significant radical removal efficiency [32].

*Blackcurrant* is present in the mixture as a dry blackcurrant extract of *Ribes nigrum* leaves, titrated at 4% in rutin. *Blackcurrant* (*Ribes nigrum* L.) is a shrub species belonging to the *Grossulariaceae* family that is widely cultivated in Europe, New Zealand, China and Australia [33]. *Blackcurrants* are a good source of polyphenols, particularly anthocyanins, phenolic acid derivatives, flavonoids and proanthocyanidins [34]. Flavonoids have several anti-inflammatory properties, some of which are believed to affect the function of the immune system; in fact, by evaluating the effect of rutin and its antihistamine effect, it was found to cause a reduction in IgE-mediated histamine release and the release of pro-inflammatory cytokines from mast cells [35]. In addition, Ashigai et al. (2018) demonstrated that dietary blackcurrant fiber has an effect on atopic dermatitis: serum IgE levels in the administration groups decreased compared to the control group in a concentration-dependent manner and induction in IFN-γ gene transcription in the spleen occurred [36].

In the mixture, *Helichrysum* was present as an extract of the flowering top of *Helichrysum italicum*, which was obtained by extraction in water with a ¼ E/D ratio. *Helichrysum italicum* is an evergreen plant native to the Mediterranean area with traditional therapeutic applications in inflammatory and allergic conditions, such as asthma and inflammatory skin conditions [37]. The flavonoids present in *Helichrysum italicum* have demonstrated inhibitory action in several pathways of the metabolism of arachidonic acid and other pro-inflammatory mediators [38], and the essential oils of *Helichrysum italicum* have been found to be useful in the healing of asthma, hay fever and eczema due to its anti-allergic properties [37,39].

In the mixture, a dry extract of *Perilla frutescensor* leaves was present, titrated at 2.5% total polyphenols. *Perilla frutescens* is an annual herb of the *Labiaceae* family which has several active components, including alkaloids, polyphenols and flavonoids, and due to the diversity of its active ingredients, *Perilla frutescens* has a wide range of pharmacological effects, including anti-allergic and anti-inflammatory effects [40]. In particular, the 8-hydroxy-5,7-dimethoxyflavanone in *P. frutescens* leaves can be used to prevent IgE-driven type I hypersensitivity reactions [41]. In addition, the aqueous extract of *Perilla frutescens* has demonstrated a systemic, local anti-allergic effect and the dose-dependent reduction in plasma histamine in vivo and in vitro [42].

A dry extract of the aerial parts of *Tanacetum parthenium* is present in the tested mixture, titrated at 0.5% parthenolides. *Parthenium (Tanacetum parthenium* L.) is a perennial plant belonging to the *Asteraceae* family. In *Parthenium*, the most important biologically active principles are the sesquiterpene lactones, the main one being parthenolide, which is found in the superficial leaf glands. This plant has anti-inflammatory and anti-allergic effects [43]; parthenolide binds to and inhibits the IκB kinase complex (IKK)β, which plays an important role in pro-inflammatory cytokine-mediated signaling [44]. Furthermore, in a previous study by Hayes et al. (1987), an extract of *Tanacetum parthenium* produced a dose-dependent inhibition of histamine release in rat mast cells [45].

Probiotics have positive effects on immune function and have proven useful in the treatment of allergies, showing the ability to restore intestinal microbial balance and modulating immune cell activation [46]. They also improve the barrier function of the intestinal mucosa and the enzymatic degradation of dietary antigens by probiotic enzymes [47]. After all, probiotics help to regulate allergic hypersensitivity reactions by suppressing the Th2-mediated response and enhancing Treg-mediated immune responses [48]: during allergic reactions, activation of TLR-9 receptors may interrupt the immune response by inhibiting Th1 cells, thus potentially counteracting the Th2-type immune response associated with allergies. Furthermore, immunosuppressive motifs can inhibit and inactivate dendritic cells, also promoting Treg conversion, which is crucial for triggering allergic cascades [49]. Several *Lactobacillus* spp. strains are known to play an important role in mediating the immune response due to the appearance of specific DNA suppressive motifs involved in immune stimulation [50]. Ishida et al., in a randomized double-blind versus placebo study, determined that the administration of *Lactobacillus acidophilus* achieved an improvement in symptoms in adult patients with perennial allergic rhinitis, but not a decrease in IgE. The results of this study indicate that lactobacilli can improve type I allergy symptoms not only in infants with an immature immune system but also in adults with fully developed immune systems [51]. In addition, the administration of *L. acidophilus* PBS066 in a bacterial mix with *L. rhamnosus* LRH020, *B. breve* BB077 and *B. longum subsp. longum* BLG240 resulted in an improvement in symptoms and quality of life of patients with allergic rhinitis and favored the development of a gut microbiota negatively associated with allergic diseases and with anti-inflammatory properties [52]. *Bifidobacterium animalis subsp. lactis* is a Gram-positive lactic bacterium belonging to the phylum *Actinobacteria* [53]. In a murine model of allergic asthma, oral administration of *Bifidobacterium animalis* improved the infiltration of perivascular and peribronchial inflammatory cells into lung sections and also resulted in a reduction in serum levels of specific immunoglobulin E, contributing to the improvement of allergic asthma [54].

The aim of this study was to confirm the anti-allergic and anti-inflammatory effects of the mixture, particularly through the combination of an in vitro effect and skin reaction to the prick test with histamine, which is the expression of mast cell degranulation. During the development phase, starting from the literature analysis, the concentration of the ingredients was considered in order to introduce, in a balanced manner, all the components of interest within the mixture. In our project to develop a new natural mixture with an anti-allergic and anti-inflammatory effect based on Quercetin, *Perilla frutescens*, *Boswellia serrata*, *Blackcurrant, Parthenium*, *Helichrysum*, *Lactobacillus acidophilus* and *Bifidobacterium animalis*, we undertook pre-clinical and clinical validation to determine whether such a product had plausible efficacy for use in interventional clinical trials in patients with allergic symptoms.

## 2. Materials and Methods

### 2.1. In Vitro Test on RBL-2H3 Cells

#### 2.1.1. Cell Line and Culture Condition

The anti-allergic activity of the tested sample was evaluated by measuring its ability to prevent degranulation processes in a rat leukemia cell line RBL-2H3 (ATCC CRL-2256), which is widely used as a mast cell simulant model in immunology and allergology studies. This basophil is from peripheral blood of a *Rattus norvegicus* with basophilic leukemia. The test was conducted on RBL-2H3 cells cultured in DMEM (Dulbecco’s modified Eagle medium) containing 10% fetal bovine serum (FBS) and 1% antibiotics (penicillin and streptomycin) and incubated under standard culture conditions (37 °C, 5% CO_2_). RBL-2H3 cells are widely used in allergy and immunology research, as they, like mast cells, express FCRI and undergo degranulation following cross-linking between FCRI-bound IgE and allergens. In this, they therefore mimic the behavior of mast cells and basophils in their response to immunological stimuli. The first step in the validation process was pre-clinical in order to assess whether the intake of the mixture was able to interfere with cell degranulation processes and the release of mediators such as histamine or histamine-like substances. The scientific protocols of the in vitro test were assessed, performed, and approved by the Technical Scientific Committee (TSC) of Bio Basic Europe S.r.l. (Milano, Italy) [55,56].

#### 2.1.2. Sample Preparation

The test sample (M) for the in vitro test was composed of active ingredients contained in 15 tablets. This was prepared because a minimum quantity of powder greater than 10 g was required for solubilization and subsequent dilutions. The sample M used in the in vitro test is a mixture of botanical extracts without excipients, tested in blend form, not subjected to further processing. The mixture contained the following: Boswellia serrata extract (Akbamax^®^), 1500 mg, purchased from Arjuna Natural Extracts Ltd. (Alwaye, India) *Blackcurrant* 4% Rutin, 1500 mg, purchased from Nutraceutica S.r.l. (Ozzano dell’Emilia, Italy) *Quercetin granular* 95%, 3150 mg, purchased from Farezis Pharma S.r.l. (Milano, Italy) Everlasting ¼ (dried extract of Helichrysum 1/4 flowering tops), 750 mg, purchased from Nutraceutica S.r.l. *Perilla frutescens* extract (dry extract of seed perilla 2.5% polyphenols), 3000 mg, purchased from Nutratrade S.r.l. *Parthenium* 0.5% parthenolide flowering tops e.s., 1500 mg, purchased from Nutraceutica S.r.l. *Lactobacillus acidophilus* (PBS066/LA001-DSM 24936/LMG P-29512) SGL11 150 mld, 199.5 mg (15 mld UFC), purchased from SynBalance S.r.l. (Lombardia, Italy) *Bifidobacterium animalis* ssp. lactis Bi1 200 mld, 150 mg (15 mld UFC), purchased from Centro Sperimentale del Latte S.r.l. The sample M was soluble in DMSO at the concentration of 2.0 mg/mL. To avoid the interference of the solvent with the results of the assay, the starting solution was diluted 100 times in complete culture medium or in the buffer used for the assay. Therefore, the cells were treated with the sample at the concentration of 0.02 mg/mL and subsequent 1:2 dilutions (in culture medium/buffer).

It was verified that DMSO had no effect at the dilutions tested. For efficacy tests, the starting solution was always diluted in cell culture medium to the final concentrations to be tested.

#### 2.1.3. Evaluation of the Absence of Cytotoxicity of the Mixture

In order to choose the concentrations of the product to use for the anti-allergic and anti-inflammatory assay, we performed a preliminary cell viability assay. The cell viability was evaluated on RBL-2H3 through a MTT test [3-(4,5-dimethylthiazol-2-yl)-2,5-diphenyltetrazolium bromide]; there was a yellow compound that was bioreduced by cells into a purple-colored formazan product. This conversion was accomplished by NADPH or NADH produced by dehydrogenase enzymes in metabolically active cells. The cells, which were seeded at a density of 2 × 10^5^ cells/well, were treated with MTT (1 mg/mL) and incubated for 3 h under standard culture conditions. After this period, the solution of MTT was discarded and 100 µL of isopropanol was added in each well in order to dissolve the formazan crystals. The absorbance (optical density, OD) was determined spectrophotometrically at a wavelength of 570 nm [57].

#### 2.1.4. Evaluation of Degranulation

RBL-2H3 cells were sensitized with anti-dinitrophenyl immunoglobulin E (anti-DNP IgE, 200 ng/mL overnight) and then stimulated with dinitrophenyl human albumin (DNP-HSA, 100 ng/mL for 60 min). One set of cells was neither sensitized nor stimulated (NC, not sensitized and not stimulated cells), a second set of cells was sensitized and stimulated (PC, positive control); a third set of cells was pre-treated with the tested sample, sensitized and stimulated (TS cells treated with the tested sample); TS cells were treated with the sample from a concentration of 0.02 mg/mL and subsequent 1:2 dilutions (always in culture medium/buffer). Dehydrocostolactone 20 μM, a sesquiterpene lactone of natural origin contained in *Saussurea Costus*, with known anti-inflammatory and anti-allergic activities, was used as a standard control (SC) [58]. The medium was then recovered and separated from the cells and incubated with 4-nitrophenyl-N-acetyl-D-glucosamide. The reaction was stopped with carbonate buffer (0.05 M NaHCO_3_/0.05 M Na_2_CO_3_) at a pH of 10. The absorbance was measured by spectrophotometric reading at 405 nm. Degranulation was calculated as a percentage of β-hexosaminidase with respect to not sensitized (NC) cells as follows:



$$Degranulation\left(\%\right)=\frac{{OD}_{X}}{{OD}_{NC}}\cdot 100$$



The release of β-hexosaminidase positively correlated with the release of histamine, which is a major component of mast cell granules; therefore, the release of β-hexosaminidase was used as a marker of mast cell degranulation.

#### 2.1.5. Dosage of TNFα

After the evaluation of the degranulation, the supernatant recovered from the cells was also used for the dosage of TNFα, which was performed by ELISA (Elisa kit, Thermo Fisher Scientific, Waltham, MA, USA) (enzyme-linked immunosorbent assay), a plate-based assay technique designed for detecting and quantifying substances such as peptides, proteins, antibodies and hormones [59]. In an ELISA assay, a specific antigen for the TNFα to identify was immobilized on a solid surface (the bottom of a well), to which the culture medium for the dosage was added. Detection was performed by means of a secondary biotinylated antibody which then reacted with streptavidin-HRP. The colorimetric reaction was proportional to the amount of TNFα present in the medium. The results were read using a spectrophotometer at 450 nm. The values obtained were then interpolated in a standard curve of TNFα.

### 2.2. Preliminary In Vivo Test

The clinical test was conducted at the otorhinolaryngology clinic of Policlinico Umberto I, Rome and promoted by La Sapienza University of Rome (Departement Committee approval for the study: University la Sapienza, Department of Oral and Maxillofacial Sciences-Prot. n. 0000442). The in vivo clinical study was conducted by monitoring changes in the histamine wheal in subjects who were taking the product.

All patients presented to the allergology laboratory with rhinitic symptoms but without an ongoing antihistamine intake. In addition, all patients gave oral and informed consent in the context of normal diagnostic practice. This allowed for the observation of the antihistaminic power of the product in relation to its interference with the development of the wheal generated following a prick test with 10 mL of histamine hydrochloride [60]. The antihistaminic effectiveness of the product was tested on nine subjects with suspected allergic rhinitis, seven males and two females between 18 and 50 years old, who took the formulation in the form of a gastro-resistant tablet once (four subjects) or twice (five subjects) a day, after the main meals. The tested mixture is shown in Table 1. The excipients included in the formula were inert substances that did not contribute to the clinical efficacy data and were added only to obtain the finished pharmaceutical form (tablet), such as binders to give compactness and volume and anti-caking agents to improve the flowability of the powders and lubricants. In order to test the allergenic reactivity of an individual patient, a prick test with a mixture of histamine hydrochloride at a concentration of 10 mg/mL was used as a positive control. The prick test with histamine hydrochloride was an expression of in vivo mast cell reactivity of an individual patient, and interference with this type of reaction is typical of all products with an antihistamine effect. We evaluated the antihistamine efficacy of the mixture under study by assessing its interference with the development of the wheal in question. The anti-allergic efficacy of this mixture was evaluated by administering to the patient this nutraceutical blend in the form of tablets, and we evaluated the wheal at three times: time zero (T0) and after 15 (T1) and 30 (T2) days. The evaluation was performed by measuring the size of the wheal that developed after a histamine prick test at the three time and subsequent comparison of the size of the wheal formed at T0 and afterwards. The results were statistically analyzed by Wilcoxon and Mann–Whitney tests [60,61]. The interference (decrease) in the size of the histamine wheal correlated with the antihistamine effect and thus with the anti-allergic effect of the mixture tested. The test is preliminary because the number of subjects is small, but it is significant for the purpose of justifying the in vitro experimental phase.

## 3. Results

### 3.1. Preliminary Cell Viability Evaluation

The absorbance (OD) measured at 570 nm was proportional to cell viability. Percentages in Table 2 were calculated using the absorbance values at 570 nm and considering 100% of the OD of the positive control (PC, untreated cells). TS = cells treated with the tested sample. According to the obtained results, we decided to perform the anti-allergic and anti-inflammatory test using concentrations of 0.02, 0.01, and 0.005 mg/mL. These were the three highest non-cytotoxic concentrations.

### 3.2. In Vitro Test Results

#### 3.2.1. Degranulation Evaluation

The mixture under examination, composed of Quercetin, *Perilla frutescens*, *Boswellia Serrata*, *Blackcurrant*, *Parthenium*, *Helichrysum*, *Lactobacillus acidophilus* and *Bifidobacterium animalis*, significantly reduced degranulation at the tested concentrations of 0.02 and 0.01 mg/mL, a reduction of 35.1% and 12.1%, respectively (Figure 1 and Table 3). This indicated its potential anti-allergic activity. The experiment clearly demonstrated the antihistaminic effect of the innovative nutraceutical blend, and as such, the next step was clinical.

#### 3.2.2. Dosage of TNFα

The mixture under examination, composed of Quercetin, *Perilla frutescens*, *Boswellia serrata*, *Blackcurrant*, *Parthenium*, *Helichrysum*, *Lactobacillus acidophilus* and *Bifidobacterium animalis*, significantly reduced TNFα at the tested concentrations of 0.02 and 0.01 mg/mL (protective activity of 13.0% and 11.9%, respectively, with respect to untreated cells). The experiment clearly demonstrated the anti-inflammatory effect of this mixture, as shown in Figure 2 and Table 4. The “protection” was calculated as the percentage reduction in TNF-α in the presence of the sample. The applied formula is as follows:
$$\mathrm{Protection}\%=100-\frac{\left[TNFaTS\right]*100}{\left[TNFaPC\right]}$$

[TNFaTS] = TNFα concentration in presence of the tested.

**Figure 2 cimb-47-00965-f002:**
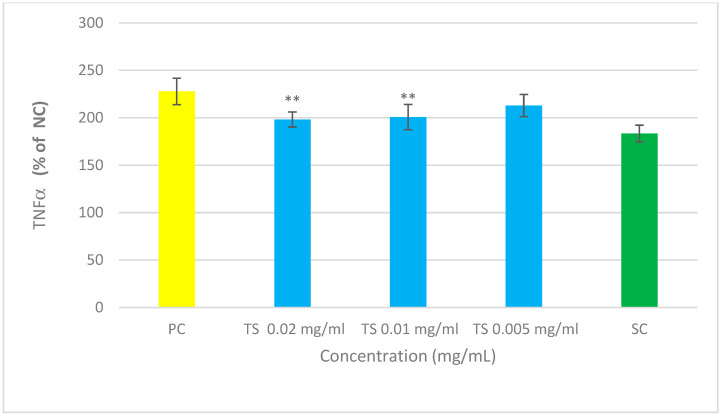
The absorbance measured at 450 nm was directly proportional to the amount of TNFα produced by the tissues. To calculate TNFα concentrations, OD values obtained were interpolated on a standard TNFα curve. Percentages were calculated relative to non-sensitized and unstimulated cells (NC). Values were expressed as means ± standard deviation. Statistical analysis of the data was performed using Student’s *t*-test. *p*-values < 0.05 were considered significant. ** *p* < 0.01 vs. PC.

**Table 4 cimb-47-00965-t004:** Percentage of protection and release of TNFα by different set of cells tested. Statistical analysis was Student’s *t*-test. *p*-values. TS = cells treated with the tested sample. SC = standard control (20 μM dehydrocostuslactone). PC = positive control (sensitized, stimulated and untreated cells).

	**PC**	**TS** **0.02 mg/mL**	**TS** **0.01 mg/mL**	**TS** **0.005 mg/mL**	**SC**
TNFα (pg/mL)	248.8	216.5	219.2	232.5	200.3
Protection (% of PC)	–	13.0	11.9	6.5	19.5

### 3.3. Preliminary In Vivo Test Results

The measurement of the size of the histamine wheals at the selected times and the application of statistical analysis according to the Wilcoxon test showed a significant difference in the size of the histamine wheals between both times T0 and T1 (*p* = 0.1; z = −2.25) and between times T0 and T2 (*p* = 0.4; z = −2.02). In contrast, there was no statistically significant difference in the size of the histamine wheals between times T1 and T2, although there was a further, albeit smaller, measured decrease in the size of the histamine wheals (*p* = 0.22; z = −1.21).

Verification for the existence of a significantly different result between subjects who took the invented mixture one or two times a day was performed by means of the Mann–Whitney statistical test. Treatment with the mixture resulted in a significant decrease in the size of the histamine wheal after 15 days (T1), with statistically significant differences from T0. These differences from T0 were also maintained at 30 days of therapy (T2). Furthermore, a comparison of the results obtained with single (one for each day) or repeated administration treatment showed that even single administration was sufficient to guarantee the antihistamine effect and that this effect was achieved in only 15 days (T1) [61].

## 4. Discussion

Some of the most commonly used drugs for the treatment of respiratory and cutaneous allergies (rhinitis and dermatitis) are antihistamines. These medications are highly effective in controlling symptoms, but over the years they have been shown to produce side effects that can negatively interfere with the quality of life of patients who take them. Respiratory allergic diseases are among the most prevalent internationally diseases; rhino conjunctivitis and asthma significantly interfere with the quality of patients suffering from them. World Health Organization (WHO) has tried to pragmatize the diagnostic and therapeutic approach to these conditions through two international documents: the ARIA and GINA documents [12,62].

The treatments for these conditions involve a proper allergological diagnosis, and when the allergen has been identified, all preventive environmental measures suggest that patients decrease contact with the allergen. Along with this, drugs have been suggested that can remedy the clinical situation; among those for upper tract allergic pathologies, antihistamines are the most widely used drugs internationally [2,12]. These products, as a limiting factor in their use, certainly have side effects in adults and children. Among the most important side effects (which are reduced in newer generation products) are those in the central nervous system (CNS), those in the gastro-intestinal tract, and, rarely, those in the heart [17,18]. Due to tolerance, in the treatment of IgE-mediated allergic diseases, patients may not respond to first-line treatment and in these patients, guidelines generally recommend increasing the dose of the antihistamine initially; however, there are side effects associated with high doses. In this case, the use of a natural product with antihistaminic effects could replace the excessive increase in the dose of the initial product through a synergistic effect, thus also avoiding dose-dependent side effects.

Despite the important efforts to develop innovative products with therapeutic results and negligible side effects, to date there are no antihistamines with these characteristics. Based on the need to create a natural product without side effects and with a plausible action, we developed a mixture with an antihistamine and anti-inflammatory effect. We started from ingredients with known anti-allergic and anti-inflammatory effects. We selected the dosages of ingredients for the composition of the mixture and for the tablet based on market analysis, maximum permitted limits and ingredient synergism; adjusting the ratios based on dosages, laboratory tests and the capacity of the chosen pharmaceutical form.

After the formulation, the aim of the study was a plan articulated in two steps: an in vitro and in vivo test. The aim of the study was to analyze a nutraceutical mixture based on Quercetin, *Perilla frutescens*, *Boswellia serrata*, *Blackcurrant*, *Parthenium*, *Helichrysum*, *Lactobacillus acidophilus* and *Bifidobacterium animalis*. We used a pre-clinical approach (in vitro) followed by preliminary clinical tests (in vivo), all for the purpose of creating a model by evaluating the degranulation-inhibiting effect on mast cell-like cells and the interference on histamine wheals during skin prick tests.

Relative to the in vitro study, it was found that the mixture of Quercetin, *Perilla frutescens*, *Boswellia Serrata*, *Blackcurrant*, *Parthenium*, *Helichrysum*, *Lactobacillus acidophilus* and *Bifidobacterium animalis* had biological properties characterized by interference with the release of a histamine-like substance by mast cell-like cells and the release of a pro-inflammatory cytokine.

During the in vitro test, statistically significant differences at the concentrations of 0.02 mg/mL and 0.01 mg/mL of sample M (extracts without excipients) were detected (reduction of 35.1% and 12.7%, respectively). The tested product was also proven to significantly reduce TNFα at the same concentrations (protective activity of 13.0% and 11.9%).

The preliminary in vivo study showed that the mixture had an ability to interfere with skin reactivity through interference with the diameter of the histamine wheal during prick tests. Furthermore, a comparison of the results obtained with single-dose (one per day) or repeated treatment showed that even single administration was sufficient to ensure the antihistaminic effect and that this effect was achieved in just 15 days (T1).

## 5. Conclusions

Therapy with antihistamines is a milestone in the management of allergic diseases, especially in the early stage of allergic reaction. Unfortunately, even new-generation products are not free of side effects, especially at higher doses, the most annoying of which are at the CNS level. In order to explore the possibility of producing a natural product with antihistaminic and anti-inflammatory effect and a solid scientific profile, we set up a validation process through an initial in vitro step on mast cell-like cells, followed by a preliminary in vivo study. The first part of the study confirmed that the investigated product was able to interfere with both the release of histamine-like mediators (β-hexosaminidase) and anti-inflammatory cytokines such as TNF alpha; the second part of the study assessed the interference of this innovative mixture with mast cell reactivity calculated through the size of the histamine wheal. The aim of the preliminary in vivo study was not to test the clinical effect of the mixture but instead the variations in millimeters of the wheal area after the administration of the mixture. The preliminary results of the in vitro and in vivo tests proved that the nutraceutical blend of Quercetin, *Perilla frutescens*, *Boswellia Serrata*, *Blackcurrant*, *Parthenium*, *Helichrysum*, *Lactobacillus acidophilus* and *Bifidobacterium animalis* may have an antihistamine and anti-inflammatory effect. Its activities and their possible evolution can be studied in the future on a larger in vivo sample and in clinical trials on pathologies, such as rhinitis and allergic dermatitis, to validate these initial but encouraging results; these future studies are also supported by the safety profiles of the nutraceutical blend studied by us, which can open the doors to innovative therapeutic strategies.

### Limits of the Study

The in vivo preliminary study was conducted with a limited sample size; we plan to expand the sample soon in order to obtain more reliable results compared to the in vitro data.

## Figures and Tables

**Figure 1 cimb-47-00965-f001:**
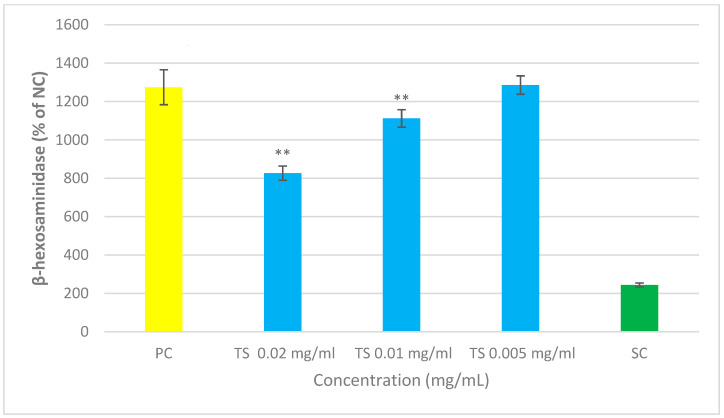
The measured OD was directly proportional to the amount of β-hexosaminidase in the analyzed fluids, which in turn positively correlated with the levels of histamine released during the degranulation phase and is therefore an indicative marker of the degranulation process. The percentages were calculated based on the measured absorbance values and represented the percentage of β-hexosaminidase compared to non-sensitized and unstimulated cells (NC). Values were expressed as means ± standard deviation. Statistical analysis of the data was performed using Student’s *t*-test. *p*-values < 0.05 were considered significant. PC = positive control (sensitized, stimulated and untreated cells); TS = cells treated with the tested sample; SC = standard control (20 μM dehydrocostus lactone; ** *p* < 0.01 vs. PC.

**Table 1 cimb-47-00965-t001:** Composition of the mixture: qualitative/quantitative composition of the mixture tested in vivo study.

Component	Quantity (mg)	Specifics
Dry gum-resin extract 75% boswellic acids, 10% keto-boswellic acid	100	75 mg boswellic acids, 10 mg keto-boswellic acid
*Blackcurrant* dry extract from leaves 4% rutin	100	4 mg rutin
Dried extract of *helichrysum* from the flowering tops	50	
Dry extract of seed *perilla* 2.5% polyphenols	200	5 mg polyphenol
Granular Quercetin (95%)	210	200 mg quercetin
*Parthenium* dry extract 0.5 parthenolides	100	0.5 mg parthenolides
*Lactobacillus acidophilus* SGL11 150 × 10^9^ UFC	13.3	1 billion CFU of live cells
*Bifidobacterium animalis* ssp. *lactis* Bi1 200 × 10^9^ UFC	10	1 billion CFU of live cells
Microcrystalline cellulose	176.7	
Silicon dioxide	20	
Magnesium salts of fatty acids	20	
Total	1000	

**Table 2 cimb-47-00965-t002:** Absorbance and cell viability at different concentrations of mixture.

	PC	0.0002 [TS] (mg/mL)	0.0003 [TS] (mg/mL)	0.0006 [TS] (mg/mL)	0.0013 [TS] (mg/mL)	0.003 [TS] (mg/mL)	0.01 [TS] (mg/mL)	0.015 [TS] (mg/mL)	0.02 [TS] (mg/mL)
OD mean	0.484	0.507	0.530	0.522	0.529	0.531	0.489	0.482	0.486
Viability (%)	100.0	104.7	109.5	107.8	109.3	109.7	101.0	99.6	100.3

**Table 3 cimb-47-00965-t003:** Percentage change in degranulation. Statistical analysis: Student’s *t*-test. PC = positive control (sensitized, stimulated and untreated cells). TS = cells treated with the tested sample; SC = standard control (20 μM dehydrocostuslactone).

	PC	TS 0.02 mg/mL	TS 0.01 mg/mL	TS 0.005 mg/mL	SC
Protection (% of PC)	–	35.1	12.7	–	80.8

## Data Availability

The original contributions presented in this study are included in the article. Further inquiries can be directed to the corresponding author.

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
