# Peer review of "Evaluation of the Antihistamine and Anti-Inflammatory Effects of a Nutraceutical Blend Based on Quercetin, Perilla frutescens, Boswellia serrata, Blackcurrant, Parthenium, Helichrysum, Lactobacillus acidophilus and Bifidobacterium animalis Through In Vitro and In Vivo Approaches—Preliminary Data"

_cimb, 2025, doi:10.3390/cimb47110965_

Round 1

Reviewer 1 Report

Comments and Suggestions for Authors

Comments for the Authors

  1. Title
    • Scientific names should be in italics (e.g., Lactobacillus acidophilus, Bifidobacterium animalis, Boswellia serrata).
    • Ensure proper capitalization (e.g., animalis instead of Animalis).
    • Terms such as in vitro and in vivo should also be italicized.
  2. Abstract
    • References should not be included in the abstract.
    • Correct formatting issues noted in the title also apply throughout the abstract.
  3. Introduction
    • References must be written in continuous ranges. For example, “[40–41–42]” should be “[40–42],” and “[28–29–30–31–32]” should be “[28–32].” Please revise consistently throughout the manuscript.
  4. Figures and Tables
    • Figure 1: The column chart should be improved, and standard deviations (SD) must be added. The x-axis should be labeled as “Concentration (mg/mL).”
    • If a column chart is provided, the corresponding table (e.g., Table 3) becomes redundant. The same applies to Figure 2 and Table 4.
    • Table headings should be placed above the tables, not below.

**Please revise the manuscript thoroughly, improve its overall quality, and resubmit it for consideration.

Author Response

Dear Reviewer, 

Best Regards

Reviewer 2 Report

Comments and Suggestions for Authors

1. Small sample size and lack of control group
The in vivo experiment included only 9 subjects (7 males, 2 females), which is a limited sample size that affects statistical power. It is strongly recommended to increase the sample size (e.g., ≥30 subjects) and include a placebo control group to rule out psychological effects or influences from the natural course of the disease.

2. Unclear dose-response relationship
The in vitro experiment tested only three concentrations (0.02, 0.01, 0.005 mg/ml). It is necessary to include a wider range of gradient concentrations (e.g., from 0.001 to 0.1 mg/ml) to establish a clear dose-response relationship.

Furthermore, the in vivo experiment did not show a significant difference between single and double daily doses. Therefore, including distinct dose groups is essential to clarify whether a dose-response relationship exists.

3. Lack of verification for probiotic viability and activity
The study did not assess the viability (CFU count) of the probiotics or their colonization effects in the gut. It is recommended to supplement the analysis with data from fecal microbiota analysis or measurements of short-chain fatty acids (SCFAs) to evaluate probiotic activity and potential gut modulation.

4. Insufficient mechanistic investigation
Although individual components are supported by literature, the study lacks investigation into the relative contribution (e.g., weight of effect) of each component (such as Quercetin, Boswellic acids, or probiotics) within the mixture. The potential synergistic or antagonistic interactions between the components in the blend were not explored. Additionally, Table 1, which is cited in the text for the mixture's composition, was not included in the provided manuscript file.

5. Other issues

â‘  Citations should not appear in the abstract; references should start from the Introduction section.

â‘¡ Reference citation style needs consistency and correction (e.g., "[1][4]" should be "[1,4]"; "[46-47-48-49-50-51-52-53-54]" should be "[46-54]").

â‘¢ Figures 1 and 2 require correction of decimal separators (e.g., "0,01" should be "0.01") and clear indication of statistical significance markers (e.g., "**") on the relevant bars/data points.

â‘£ Tables 3 and 4 lack descriptive titles. The title and content headings of Table 2 are unclear.

⑤ The reference list format has significant inconsistencies and does not appear to conform to the CIMB journal's style. A thorough check and correction according to the journal's guidelines are imperative. References should also be cited based on necessity; excessive or irrelevant citations should be avoided.

Author Response

Dear Reviewer, 

Best Regards

Reviewer 3 Report

Comments and Suggestions for Authors

This manuscript by Simonetta et al. investigates the antihistaminic and anti-inflammatory effects of a natural compound mixture containing Quercetin, Perilla frutescens, Boswellia serrata, and Blackcurrant, among others, using both in vitro (RBL-2H3 cell model) and in vivo (histamine prick test in humans) approaches. The authors report modest effects, including reductions in mast cell degranulation (up to 35% in vitro), histamine wheal size (up to 30% in vivo), and TNF-α levels (13% in vitro). The study addresses a timely and relevant topic, given the growing interest in natural alternatives to synthetic antihistamines with fewer side effects. The introduction provides a solid contextual background, and the discussion appropriately relates the findings to the known bioactivities of the mixture’s components. However, the work is limited by methodological weaknesses and overinterpretation of preliminary data, which diminishes confidence in the conclusions. Overall, the findings are preliminary and intriguing but require major revisions to improve scientific rigor.

Major concerns:

  1. No power calculation is provided for either the in vitro or in vivo experiments, raising concerns about whether the study was sufficiently powered to detect meaningful effects.
  2. The human study includes only 9 participants (7 males, 2 females) with suspected allergic rhinitis—an insufficient sample size given inter-individual variability. The absence of a placebo group, randomization, and blinding further undermines the validity of the findings. A properly powered, randomized, double-blind, placebo-controlled design (≥20–30 participants per group) is strongly recommended. Please also clarify inclusion/exclusion criteria (e.g., age, allergy severity, concurrent medications).
  3. The tested concentrations (0.005–0.02 mg/ml) produce dose-dependent effects, but their physiological relevance is unclear. The authors should discuss expected plasma or tissue concentrations following oral ingestion and provide pharmacokinetic data or supporting references to justify in vivo extrapolation.
  4. The composition listed for the in vitro tests (Section 2.2.3) differs from that of the in vivo tablets (Table 1). Please clarify whether these are equivalent formulations and describe standardization procedures, including extraction methods and batch variability.
  5. Please provide the number of replicates for each experiment and add error bars to the plots.
  6. Without orthogonal data or complementary assays (such as binding studies, or functional validation by mutational analysis), some of the mechanistic interpretations are purely speculative. Please include a section describing the limitations of this study, and tone down the language stating the conclusions.
  7. I have concerns regarding the “oral and informed consent”. Can the authors expand on how they define “informed consent”?
  8. Another concern relates to the lack of demonstrated novelty of the mixture. In the introduction, the authors note that “this selection was not occasional and each ingredient is supported by extensive scientific literature.” While the manuscript provides background on the individual components, it does not explain what distinguishes this specific combination from existing formulations. The potential synergistic or additive molecular effects arising from combining these ingredients; presumably the central premise of the study, are not discussed or experimentally addressed.

Author Response

Dear Reviewer, 

Best Regards

Round 2

Reviewer 1 Report

Comments and Suggestions for Authors

Comments

  • The manuscript presents an interesting preliminary investigation into the antihistamine and anti-inflammatory effects of a complex mixture containing plant extracts and probiotics. The topic is relevant and the introduction provides a solid background on allergy mechanisms and current therapeutic approaches. However, several methodological and structural issues should be addressed to improve clarity, reproducibility, and scientific rigor.

  • Introduction
  • The introduction provides a good overview of allergy pathophysiology and current treatments.
  • In the last paragraph, the authors state that the selection of plants and probiotics was based on previous scientific literature. It would be more appropriate to cite this supporting literature directly in the introduction rather than deferring it to the Discussion section. This will strengthen the rationale for the study and help readers understand the basis for choosing these specific ingredients.

  • Materials and Methods

Section 2.2.3 — Sample M

  • The description of Sample M needs clarification. It appears that the materials used are dietary supplements, but the manuscript does not specify whether they were in capsule, syrup, or another form.
  • For example, Boswellia serrata extract (Akbamax® 1500 mg) is a capsule product that also contains excipients besides the active extract. Please clarify whether you used the capsule content directly or a purified extract.
  • Mention that in the title. “dietary supplements”

Section 2.2.2 — Sample Preparation

  • This section needs significant revision for clarity. The solvent and final concentrations used for each assay are not clearly described.
  • Instead of a separate “Sample Preparation” section, it would be clearer to include the concentration and solvent details within each experimental assay subsection.

Section 2.2.3 — Sample M (redundancy)

  • This section could be merged with the Materials Section, where all products, compounds, and solvents should be listed together for better organization.

Section 2.2.4 (line 187)

  • Please specify the pH and concentration of the carbonate buffer used to stop the reaction in the β-hexosaminidase assay.

Table 1 — Composition of the Mixture

  • Each component listed in the table should also be described in the Materials section, including manufacturer, form, and purity.
  • Parthenium is mentioned in Table 1 but not described in Section 2.2.3. Please ensure consistency.
  • Additionally, the inclusion of excipients such as microcrystalline cellulose, silicon oxide, and magnesium salts of fatty acids should be justified in the Discussion, with supporting references explaining their relevance or functional role in the formulation.
  • Please also justify the specific quantities used for each ingredient (e.g., why 100 mg of blackcurrant was chosen instead of 50 or 200 mg). Such rationale should be provided in the Discussion section, supported by references or formulation logic.

Table 2

  • The absorbance and cell viability values appear to fall within a 10% error range, indicating no statistically significant difference between the positive control and the tested sample concentrations. The authors are requested to clarify whether these differences were found to be significant. Additionally, please explain the rationale for selecting the concentrations of 0.02, 0.01, and 0.005 mg/mL for the subsequent assays, given that no apparent differences were observed among them.

Figure 1

  • The Y-axis is labeled “% of NC,” yet the values range up to 1600%. Please verify the calculation method for normalization against the negative control and correct if necessary.
  • Ensure that axis labels and units are consistent and logically scaled.
  • Same thing for Figure 2.

Table 3

  • Please clarify how “protection” was calculated. Include the formula and reference if applicable.

Additional Suggestions

  • Since the formulation includes both plant extracts and probiotics, I recommend adding antioxidant or free radical scavenging assays (e.g., DPPH, ABTS, FRAP) to strengthen the mechanistic link to the observed biological effects.
  • It would also be valuable to test individual ingredients to identify which components contribute most strongly to the activity before combining them into the final formulation.
  • Future studies should include in vivo experiments on larger sample sizes and over longer treatment durations to confirm these preliminary findings.

Discussion

  • The Discussion section is broad but somewhat general. It would benefit from tighter focus on the key findings and stronger integration of references that support the mechanisms proposed for each component.

Conclusion

The study explores a promising multi-component natural formulation with potential antihistamine and anti-inflammatory properties. However, the Materials and Methods require clearer description, and the data interpretation should be strengthened with appropriate justification and discussion. Addressing these issues will significantly improve the quality and reproducibility of the manuscript.

Reviewer 2 Report

Comments and Suggestions for Authors

Despite the authors having amended the title to indicate a preliminary study, the following critical issues remain and should be addressed prior to any potential acceptance in CIMB. The final decision, of course, rests with the journal's editors.

  1. Small Sample Size and Lack of Control Group

    • While the authors cite Kühnast et al. (2008) to justify the use of non-parametric tests for a small sample size, they fail to provide a satisfactory explanation for why a control group could not be incorporated into the current study design, even one of a preliminary nature.

  2. Unclear Dose-Response Relationship In Vitro and In Vivo

    • The response to this critique, which relies on the preliminary nature of the study and references future work, is problematic. It does not adequately address the fundamental need for a clearer dose-response characterization. Furthermore, the issue of the lack of significant difference between the two dosing regimens in the in vivo experiment remains unresolved.

  3. Lack of Verification for Probiotic Viability and Activity

    • The authors correctly state that the RBL-2H3 cell model is unsuitable for assessing gut colonization. However, the absence of any data on probiotic viability or a mechanism-specific assay weakens the rationale for including probiotics in the natural mixture. Without such evidence, the contribution of the probiotics to the observed effects remains speculative and unjustified.

  4. Insufficient Mechanistic Investigation into Synergistic Effects

    • Although the authors presented new, unpublished data in their response letter suggesting synergistic effects, their decision not to incorporate these findings into the main manuscript is a significant weakness. Without formally including this data—complete with methodology and statistical analysis—in the results and discussion sections, the proposed mechanism of synergy remains unsubstantiated and unclear to the reader.

  5. References Formatting

    • The reference list has not been fully formatted according to the specific guidelines of CIMB. A thorough check and correction to ensure strict adherence to the journal's style are imperative.

Reviewer 3 Report

Comments and Suggestions for Authors

The authors have sufficiently addressed my concerns. I recommend publication in the present form.

Author Response

Dear Reviwer, Please see the attachment

Round 3

Reviewer 2 Report

Comments and Suggestions for Authors

The authors have made some efforts to address the comments, primarily by adjusting the title and improving formatting. However, after careful evaluation of the authors' responses and the second revision, I find that the most critical scientific concerns raised during the review process have not been adequately resolved. Consequently, I must recommend rejection/or major revision of the manuscript of cimb-3911399 in its current form. The fundamental issues that preclude publication are as follows: 1. Lack of a Proper Control Group in the In Vivo Study: The authors' justification for not including a parallel control group (e.g., placebo) is scientifically unsound. Their argument conflates the positive control used to validate the skin prick test procedure (histamine) with the essential study design element of a parallel control group. Without a control group, it is impossible to determine whether the observed reduction in wheal size is attributable to the investigational mixture or to natural variations in histamine reactivity, regression to the mean, or other external factors over time. This flaw severely undermines the validity and interpretability of the in vivo results. 2. Unsubstantiated Claim of Synergistic Effects: The authors' refusal to incorporate data on the synergistic effects of the mixture into the manuscript is a critical failure of scientific transparency. They cited ongoing patent applications as the reason. However, making a claim of synergy—which is a central and compelling aspect of their narrative—without providing the underlying data, methodology, and statistical analysis for peer review is unacceptable. It prevents the scientific community from verifying this key claim and violates a core principle of academic publishing. If the data cannot be published, the claim should not be made. 3. Insufficient Dose-Response Characterization: The authors' response that a clear dose-response relationship is not necessary due to the "broad effect" of a natural blend is not convincing. Establishing a dose-response is a cornerstone of pharmacological and nutraceutical research to demonstrate a genuine, concentration-dependent biological effect. Their failure to provide this, both in vitro and in vivo, weakens the evidence for a specific, testable effect. 4. Lack of Direct Evidence for Probiotic Contribution: While the literature supports the general role of probiotics in allergy, the authors provide no direct evidence (e.g., viability assays, mechanism-specific tests) that the specific probiotics in their blend contributed to the observed in vitro effects on mast cells. Their contribution remains entirely speculative within the context of this study.

Author Response

1-Dear Reviewer, thank you for your observation. We want to underline that the variability in the diameter of the histamine wheal is very low, instead the wheal generated by different allergens depends on the sensitization of individual patients. The aim of the study was the in vitro part and in vivo part was preliminary and our control was intra-group: pick test before the treatment.

2-We would like to emphasize that, as a referee, you have the opportunity to evaluate our statement transparently through the document sent. For reasons related to the patent, we cannot publish them. However, that this will not diminish the scientific value of the study.

3- Dear Editor, this point will be the subject of further study in a dedicated study. Thank you for your suggestion.

4-Thank you for this observation we will take your suggestions into consideration in a future project.

Thank you for your constructive interactions with our group and for your willingness to engage in scientific dialogue. We would like to emphasize that this is preliminary data, which will be followed by a double-blind, placebo-controlled clinical trial to evaluate the clinical efficacy of the product in allergic diseases.